



# Technical note: Challenges of detecting free tropospheric ozone trends in a sparsely sampled environment

Kai-Lan Chang[1,2], Owen R. Cooper[1,2], Audrey Gaudel[1,2], Irina Petropavlovskikh[1,3], Peter Effertz[1,3], Gary Morris[3], and Brian C. McDonald[2]

[1]Cooperative Institute for Research in Environmental Sciences, University of Colorado, Boulder, CO, USA
[2]NOAA Chemical Sciences Laboratory, Boulder, CO, USA
[3]NOAA Global Monitoring Laboratory, Boulder, CO, USA

**Correspondence:** Kai-Lan Chang (kai-lan.chang@noaa.gov)

**Abstract.** High quality long-term observational records are essential to ensure appropriate and reliable trend detection of tropospheric ozone. However, the necessity for maintaining high sampling frequency, in addition to continuity, is often underappreciated. A common assumption is that so long as long-term records (e.g., span of a few decades) are available, (1) the estimated trends are accurate and precise, and (2) the impact of small-scale variability (e.g., weather) can be eliminated. In

this study we show that the undercoverage bias (e.g., a type of sampling error resulting from statistical inference based on sparse or insufficient samples, such as once-per-week sampling frequency) can persistently reduce the trend accuracy of free tropospheric ozone, even if multi-decadal time series are considered. We use over 40 years of nighttime ozone observations measured at Mauna Loa, Hawaii (representative of the lower free troposphere) to make this demonstration, and quantify the bias in monthly means and trends under different sampling strategies. We also show that short-term meteorological variability

remains a cause for an inflated long-term trend uncertainty. To improve the trend precision and accuracy due to sampling bias, two remedies are proposed: (1) a data variability attribution of colocated meteorological influence can efficiently reduce estimation uncertainty and moderately reduce the impact of sparse sampling; and (2) an adaptive sampling strategy based on anomaly detection enables us to greatly reduce the sampling bias, and produce more accurate trends using fewer samples compared to an intense regular sampling strategy.

## 1   Introduction

Tropospheric ozone is the third most important greenhouse gas (after carbon dioxide and methane, Gulev et al. (2021)). Ozone is also a surface pollutant detrimental to human health and crop productivity (Fleming et al., 2018; Mills et al., 2018). The lifetime of tropospheric ozone ranges from minutes in the boundary layer to roughly three weeks in the free troposphere (Young et al., 2013), and its sources include photochemical production from various precursor gases and stratosphere-troposphere

exchange, which is challenging to accurately quantify because emissions of ozone precursor gases, atmospheric transport pathways, and extreme weather patterns also change over time (Stohl et al., 2003; Zhang et al., 2016). While the observations from remote high elevation sites can be used to quantify regional-scale ozone trends and variability within the planetary



boundary layer or lower free troposphere (Cooper et al., 2020), only sparse profiles from ozonesondes, lidars or aircraft are available to monitor ozone in the mid- or upper troposphere.

Trend detection of free tropospheric ozone at a global scale is particularly challenging because ozone is highly dynamic and observations are too limited (infrequent in time and sparse in space). In terms of long-term observations, ozonesonde and aircraft provide high quality ozone observations throughout the depth of the troposphere with a fine vertical resolution, but these programs are expensive to maintain and sampling rates are often quite low. Within the global ozonesonde network (Tarasick et al., 2019), only three sites, Hohenpeissenberg (Germany, 1966-), Payerne (Switzerland, 1968-) and Uccle (Belgium, 1969-) manage to launch ozonesondes with a sampling frequency of two or three times a week. Even so, consistent long-term free tropospheric trends cannot be found between these three western European sites in relatively close proximity (Chang et al., 2022). While most of the other ozonesonde sites target a once-per-week sampling frequency, the actual sampling rate is often less (e.g., the NOAA Global Monitoring Laboratory (GML) ozonesonde record at American Samoa has an average sampling rate of 35 profiles per year), and therefore the precision and accuracy of trends estimated from these time series might be even less reliable. Of the other instruments, only the lidar operated at the Jet Propulsion Laboratory Table Mountain Facility (California) manages to provide ozone profiles with varying frequencies of 2-5 times a week since 1999 (Chouza et al., 2019).

The IAGOS (In-Service Aircraft for a Global Observing System) program is also an important source of tropospheric ozone observations. Since 1994, IAGOS commercial aircraft have provided ozone profiles worldwide, and because the program's ozone instruments are calibrated regularly, its observational record can be considered to be a reference data set (Tarasick et al., 2019). Nevertheless, from a sampling point of view, since the availability of IAGOS data is tied to pre-determined flight schedules, the sampling schemes are often irregular and intermittent. Western Europe is the only region with abundant and near-continuous ozone measurements since 1994, with an average of more than 100 profiles per month (mostly from Frankfurt, Paris, Munich, Brussels, Dusseldorf and Amsterdam). The IAGOS observations collected at other regions often have data gaps (several years in some cases), and therefore the reliability of trend estimates based on these time series might be subject to higher levels of uncertainty.

With the exceptions discussed above, free tropospheric ozone observations are either sparse in time (once-per-week sampling) or intermittent. However, these free tropospheric vertical profiles are still the only long-term observational records to validate satellite data and global chemistry-climate model simulations. The common approach to compare satellite data to in situ or ground-based observations is made by spatially and temporally co-located comparisons between each individual profile and the corresponding satellite value (Zhang et al., 2010), or at a monthly aggregated scale (Ziemke et al., 2006). This study shows that the sampling errors (due to a sparse sampling frequency) might persistently bias the trend estimate even if multi-decadal records are considered. Therefore, the implication is that inconsistent trends can exist between ground-based and satellite observations due to different sampling schemes.

In terms of the impact of sampling on tropospheric ozone monthly means, Logan (1999) analyzed ozonesonde profile data and suggested at least 20 profiles are required to maintain the 2-sigma range below $\pm 30\%$ of the monthly means near the extratropical tropopause, and less than $\pm 15\%$ of the monthly means for tropical and extratropical free troposphere. Using the IAGOS commercial aircraft profiles above Frankfurt, Saunois et al. (2012) evaluated the sampling uncertainty on seasonal



means at 100 hPa vertical resolution, and found the uncertainty around 10% at 700-500 hPa and 15% at 400 hPa for a sampling frequency of 4 profiles per month, however, the uncertainty can be reduced to 5% and 8%, respectively, if the sampling frequency increases to 12 profiles per month. Cooper et al. (2010) merged all April-May ozone profiles (all available ozonesonde, lidar and aircraft measurements) above western North America to show that ozone had increased in the free troposphere over

1995-2008, and determined that 50 profiles per April-May season (or 25 profiles per month) are required to produce a seasonal column mean value in the 3-8 km range within $\pm 2\%$ bias.

In terms of the impact of sampling frequency on tropospheric ozone trends, by using the IAGOS profiles above western Europe where the trend values are around 1-2 ppbv/decade for 950-400 hPa and between 2-9 ppbv/decade for 350-250 hPa over 1994-2017 (50 hPa resolution), Chang et al. (2020) found that at least 10 profiles per month are required to detect the

signal at 2-sigma confidence and 18 profiles per month are required for the trend bias to be less than 5%, based on basic multiple linear regression; the requirement can be alleviated to 4 and 14 profiles per month, respectively, if a sophisticated statistical method (designed to avoid overfitting to the spurious variability at individual pressure levels) is applied (Chang et al., 2020). However, a higher sampling rate is required for a weaker signal (e.g., < 1 ppbv/decade).

As motivation for this study, Figure 1 shows the vertical profiles of ozone launched at Trinidad Head, California, from two

intensive sampling campaigns, including 30 ozonesondes launched in August 2006 and 36 ozonesondes launched in May 10 - June 19 2010 (all data links can be found in the Data Availability Section). At first glance a tumultuous and unstructured variability is revealed by the individual profiles, during both campaigns. With a focus placed on the free troposphere (700-300 hPa), individual sondes are typically characterized by irregular vertical variability, but with sufficient sampling the profile averages generally much smoother. To simulate once per week or three times per week sampling strategies, we randomly select

4 or 12 profiles from each one-month campaign to produce the subsampled mean profiles, and repeat this process 1000 times. We find that the ranges of sampling variability based on 4 samples in a month (i.e. once-per-week sampling strategy) remain very uncertain. In terms of absolute percentage deviation from the overall mean (evaluated at 10 hPa resolution layers), average deviations of 12% (August 2006) and 17% (May-June 2010) are found in the free troposphere for 4 samples; these deviations can be reduced to 6% and 9%, respectively, if 12 samples in a month are used (for a reference, average deviations between

individual sondes and the overall mean are 25% in August 2006 and 30% in May-June 2010, and an accuracy of $\pm 5\%$ is generally achieved with ozonesondes in the troposphere (Tarasick et al., 2021)). This amount of sampling variability is roughly comparable with the IAGOS data above Europe (Saunois et al., 2012).

No similar intensive daily sampling campaigns are available in tropics (e.g., at most only 14 profiles were launched from Hilo, Hawaii, in March 2001 during the TRACE-P campaign (Oltmans et al., 2004)). However, by comparing the ranges of

the 5th and 95th percentiles in the free tropospheric ozonesonde records at Trinidad Head and Hilo, we find similar variability in Jun-Jul-Aug and Sep-Oct-Nov, and a modest higher variability at Hilo in Dec-Jan-Feb and Mar-Apr-May (Figure S1, the implications for stratospheric intrusions will be discussed later). We can thus expect that the sampling issue is not less important in the tropics than in northern mid-latitudes.

To translate the impact of sampling bias on monthly means into trends, we utilize the monthly mean time series of nighttime

temperature and ozone at Mauna Loa Observatory (MLO, representative of the lower free troposphere, see Section 2 for data





descriptions) to show the sampling impact on trends in Figure 2, based on monthly means from full sampling (7 days per week), and once-per-week sampling taken just on Sunday or Tuesday (these two days of the week are chosen to represent the extreme cases, the detailed comparison of the trends from each days of the week will be discussed in Section 3, and the complete daily time series are shown in Figure S2), so each month mean is produced from ∼30 or ∼4 daily values, respectively. We can clearly

see how the sampling biases are produced due to reduced sampling. For the temperature data, even though some differences can be observed, the trend and its uncertainty are similar between the results from the full and subsampled records. The bias in the once-per-weekly sampled ozone monthly means can be very large in some cases (e.g., greater than 10 ppbv, also indicated in Figure 1), indicating that ozone is far more variable than temperature. The MLO ozone trends based on once-per-week sampling can be biased high by 29% or biased low by -46% depending on different days of the week. This figure also shows

that even though over 40 years of weekly ozone data were collected, trends can still go undetected when the samples are not representative.

Quantitatively, if we compare each ozone daily value to its monthly mean at MLO over 1980-2021, the mean absolute deviation is 20% or 8 ppbv (in Figure S1, average deviations of 19% above Trinidad Head and 25% above Hilo are found between individual sondes and their seasonal climatologies in the free troposphere, so the sampling variability is roughly

consistent between sonde and surface-based observations). Therefore, the main topic of this study can be stated as: *how many samples in a month are required to eliminate the impact of 20% interdaily variation on monthly means and trend estimates into an acceptable range?* Note that the sampling deviation associated with each daily value should not be considered to be sampling bias, the sampling bias occurs only if we have an insufficient number of samples to infer a monthly mean value or trend.

To better quantify the sampling impact, it is important to distinguish the effect of sample size and sampling bias from a sparsely sampled scheme on trend detection. Traditionally the trend analysis of atmospheric composition time series often implicitly assumes that the samples are representative and can be used to assess trends, so sample size is the major factor to determine how long the trends can be detected (given the magnitude of trend, data variability and autocorrelation are constants, Weatherhead et al. (1998)). Therefore, in order to clarify the conceptual difference, we define sample size as *the number of*

*points used to fit a statistical model (regardless of the sampling rate)*, and undercoverage bias as *a type of sampling error resulting from statistical inference based on sparse or insufficient samples*. This conceptual difference can be clearly shown in Figure 2: The three estimates of ozone trends are based on the same sample size (i.e., the same number of total monthly means), but the trend estimates from reduced samples are severely biased due to undercoverage bias. This work aims to investigate the impact of a series of biased (or non-representative) monthly means on long-term trends.

The goals of this study are (1) determine the minimum number of ozone observations necessary for accurate trend quantification in the tropical free troposphere, (2) develop optimal sampling strategies that will improve trend detection when faced with limited resources, and (3) leverage co-located observations (e.g. temperature and humidity) and climate indices (e.g., El Niño-Southern Oscillation (ENSO) and quasi biennial oscillation (QBO)) to improve trend detection through multiple linear regression techniques. Previous attempts to evaluate free tropospheric ozone trends are typically adjusted by ENSO and QBO

only (which is the standard approach for stratospheric ozone trends). We aim to show that adjustments based on co-located



meteorological observations are more pertinent to tropospheric ozone trend detection and attribution. Section 2 introduces the data sets used in this study, and discusses how the trend and its uncertainty are estimated. Section 3 conducts a thorough investigation of the impact of different sampling schemes on the bias in monthly means and trends. Section 4 provides a summary of this study.

## 2   Data and method

### 2.1   In-situ measurements

In this study we used the hourly ozone dataset measured at Mauna Loa Observatory (MLO), Hawaii (19.5°N and 155.6°W; 3,397 m above sea level (Oltmans and Komhyr, 1986; NOAA GML, 2023b)) to investigate the impact of different sampling frequencies and strategies on the estimates of monthly means and trends, with a special focus on the quantification of improvement in trend and uncertainty estimates when the sampling frequency is increased. Since MLO is located in the central North Pacific Ocean and at the northern edge of the tropics, ozone variability at MLO is impacted by mid-latitude dry air masses from the north and west, and tropical moist air masses from the south and east (Gaudel et al., 2018; Cooper et al., 2020). Lin et al. (2014) found that the relative frequency of dry and moist air mass transport from high latitudes (typically higher ozone) and low latitudes (typically lower ozone) can be influenced by short-term climate variability, such as ENSO and the Pacific Decadal Oscillation. Therefore, an adjustment for meteorological and climate variability is important for quantifying the long term-trend at MLO (Chang et al., 2021).

The MLO record is an ideal testbed for investigating free tropospheric sampling issues, because: (1) MLO is a high elevation site, and nighttime ozone at MLO is representative of the lower free troposphere (Price and Pales, 1963; Oltmans and Komhyr, 1986; Tarasick et al., 2019; Cooper et al., 2020); (2) even though some large interannual variability is present, the overall trends are roughly linear since the mid-1970s (Chang et al., 2021, 2023a), and therefore the results from different sampling tests are comparable (e.g., if the trends are strongly varying between different time periods, it will be difficult to distinguish the influence between sampling error and nonlinearity in trends). Note that since daytime data are excluded to avoid the influence from localized anthropogenic emissions (Cooper et al., 2020), no noticeable offset and heterogeneity between different days of the week, such as the ozone weekend effect, are found in the MLO nighttime ozone record (see Figure S3).

Once-per-day nighttime ozone averages (08:00-15:59 UTC) at MLO are calculated to represent lower free tropospheric ozone above Hawaii. While reliable hourly ozone observations are available at MLO for the periods 1957-1959 and 1973-1979 (Cooper et al., 2020), colocated meteorological data are more complete since the late 1970s, and therefore our study focuses on the 42-year period from 1980 to 2021. We use a 50% data coverage criterion to determine the data availability, e.g. we require 15 daily averages in a month. At the daily level, only 757 out of 15,341 (4.9%) daily ozone values were missing for the 42 years over 1980-2021. At the monthly level, 12 out of 504 (2.4%) monthly values failed to meet the 50% data coverage criterion (listed in Table S1). To avoid selecting non-representative subsamples, data from those 12 months are discarded from our sampling analysis. Links for all the data sets are provided in the Data Availability Section.



## 2.2 Statistical models for evaluating trends and uncertainties

Trend estimates are derived and compared based on the time series produced from the complete data record (full sampling) and reduced subsamples (according to different sampling strategies) at two stages: at the first stage only fundamental components for time series decomposition are considered for trend detection, i.e., a seasonal cycle and a linear trend; At the second stage

several meteorological variables and climate indices are incorporated into the model in order to improve the trend estimate, as adjustments for meteorology and short-term climate variability are often considered to be important to attribute ozone variability (Lin et al., 2014; Porter et al., 2015; Chang et al., 2021). Let $y_t, t = 1, \cdots, n$ be the ozone mean time series, each mean value is produced by an aggregation of available data collected over a temporal interval $t$ (typically on a monthly scale), the statistical models for the first and second stages can be expressed as follows, respectively:

M1 (basic model) : $y_t = \alpha_0 + \left[ \alpha_1 \sin(2\pi \frac{\text{Month}}{12}) + \alpha_2 \cos(2\pi \frac{\text{Month}}{12}) + \alpha_3 \sin(2\pi \frac{\text{Month}}{6}) + \alpha_4 \cos(2\pi \frac{\text{Month}}{6}) \right] + \beta_0 t + N_t,$

     M2 (full model) : $y_t = \alpha_0 + \left[ \alpha_1 \sin(2\pi \frac{\text{Month}}{12}) + \alpha_2 \cos(2\pi \frac{\text{Month}}{12}) + \alpha_3 \sin(2\pi \frac{\text{Month}}{6}) + \alpha_4 \cos(2\pi \frac{\text{Month}}{6}) \right] + \beta_0 t$

$$+ \beta_1 \text{WindSpeed} + \beta_2 \text{WindDirection} + \beta_3 \text{Temperature} + \beta_4 \text{RelativeHumidity} + \beta_5 \text{Dewpoint}$$

$$+ \beta_6 \text{ENSO} + \beta_7 \text{QBO}_{30\text{hPa}} + \beta_8 \text{QBO}_{50\text{hPa}} + N_t,$$

where $\alpha_0$ is the intercept, $\{\alpha_k, k = 1, \cdots, 4\}$ is a set of coefficients jointly representing the seasonal cycle, $\beta_0$ is the trend value,

$\{\beta_k, k = 1, \cdots, 8\}$ is a set of coefficients associated with different meteorological variables and climate indices, respectively, and $N_t$ is the residual series. Note that the M2 (full model) does not represent our final trend model, as a variable selection procedure will be carried out to determine which variables are the most statistically and scientifically meaningful.

     The models are fitted based on the least squares (LS) and least absolute deviations (LAD) for the estimations of mean and median trends (as well as other coefficients, albeit the focus is placed on the trend estimate), respectively. The estimations

of LAD are implemented using quantile regression available from the R package *quantreg* (Koenker and Hallock, 2001). In addition, the moving block bootstrap (MBB) algorithm is integrated into LS and LAD estimations in order to produce consistent uncertainty estimates between the mean and median trends (Fitzenberger, 1998; Lahiri, 2003), because the autocorrelation and heteroscedasticity are not invariant between different subsampled time series, an MBB approach is expected to accommodate a larger class of autocorrelation structures and be more flexible than a fixed autoregressive model, such as an AR(1) or AR(2)

process.

     The fitting procedure for LS and LAD trends is conducted iteratively and outlined as follows: (1) for each iteration, a trend model is fitted to randomly selected multi-blocks of resampled data, and the corresponding bootstrapped trend value is extracted; (2) the final trend estimate (and its 1-$\sigma$ uncertainty) is produced by the mean (and standard deviation or SD) of the bootstrapped trends from 1000 iterations. The code for implementing median regression based on the MBB algorithm is

documented in the Tropospheric Ozone Assessment Report (TOAR) statistical guidelines (Chang et al., 2023b).



In terms of fitting quality, root-mean-square (percentage) deviation and mean absolute (percentage) deviation are used to assess the overall predictive performance:

$$\text{RMSD} = \left( \frac{\sum_{t=1}^{n} (\hat{y}_t - y_t)^2}{n} \right)^{1/2} \text{ and MAD} = \frac{\sum_{t=1}^{n} |\hat{y}_t - y_t|}{n}, \text{ for the units of ppbv;}$$

$$\text{RMSPD} = \left( \frac{\sum_{t=1}^{n} ((\hat{y}_t - y_t)/y_t)^2}{n} \right)^{1/2} \text{ and MAPD} = \frac{\sum_{t=1}^{n} |(\hat{y}_t - y_t)/y_t|}{n}, \text{ for the units of percentage,}$$

where $\hat{y}_t$ is the fitted value of $y_t$. To explicitly quantify the sampling impact, by using the same methodology, we can define the undercoverage bias as $(s_r - s_c)/s_c$, where $s_c$ is the statistic of interest (e.g., can be either the monthly mean, trend value, or trend uncertainty) derived from the complete dataset, and $s_r$ is the statistic derived from the reduced or subsampled dataset, thus RMSPD and MAPD can also be used to assess the improvement due to sampling enhancement.

It should be noted that Chang et al. (2021) used an adaptive nonlinear trend technique (i.e., regression splines fitted through
generalized additive models (GAM), and no assumptions of the shape of trends is required in advance) to model the ozone variability at MLO, and found that the nonlinearity captured by the GAM is largely diminished and becomes roughly linear after the meteorological variability is accounted for, indicating that the nonlinearity in the ozone time series at MLO can be attributed to meteorological influence. Therefore, a change point analysis of long-term trends at MLO is not considered in this study.

## 3   Results

### 3.1   Quantifying representatives bias in monthly means and trends from weekly samples

Figure 3 shows the scatter plots between the monthly means based on all available (complete) daily nighttime observations and monthly means aggregated from (reduced) weekly samples, according to each day of the week and different bias exceedance thresholds (e.g., the dark red color indicates that the absolute sampling bias of the mean from reduced samples is greater
than 25% of the mean from complete samples in a given month). In contrast to RMSD and MAD which represent the overall predictive performance, bias exceedance rate is a measure focusing on the frequency of extreme sampling bias. Overall, even though we do not observe strong systematic biases from the scatter plots (as indicated by the correlation line in blue), some large discrepancies are present (see Table 1 for the results by each day of the week). On average, 13.3% and 5.2% of the months show the bias exceedance rate to be greater than 15% and 20% of the monthly means, respectively. Depending on the locations
of these discrepancies in the time series, they can obscure the true trend estimate, as shown in Figure 2.

A once-per-week sampling analysis is carried out by estimating the mean and median trends based on each day of the week and different time periods. The following discussion is based on Figure 4 (a and c):

- First of all, it is important to separate the effect of sample size (for fitting the trend model) from undercoverage bias (due to sparse sampling). The scenario on the left of Figure 4(a) indicates the mean trends and 2-sigma intervals estimated
from full sampling (7 samples per week). Trend values are around 0.9 ppbv/decade since 1980, 1990 and 2000, but the





uncertainty grows when the time periods become shorter. It implies that high certainty (2-sigma confidence level) in trends cannot be attainable over 2000-2021 and 1995-2021, mainly because the time series is too short (i.e., roughly 30 years of continuous data are required to detect the signal at this magnitude, given the trends are relatively linear over 1980-2021).

– However, given the sample size for trend estimation is the same (i.e. the total amount of monthly means for full and weekly sampling), if we sample on Tuesday or Wednesday only, high certainty in mean trends cannot be obtained even with 30 years of data (1990-2021); and if we sample on Tuesday only, high certainty in mean trends cannot be obtained even with 40 years of data (1980-2021). In these cases, we conclude that the sampling biases from weekly samples are not neutralized even when very long-term records are considered, and the undercoverage biases persistently reduce trend
accuracy.

   – So far, the above discussion is based on the mean estimator. The big picture remains similar if we compare the results between the mean and median estimators. Comparable patterns can be observed when the time series is sufficiently long ($> 30$ years), and it is not unexpected to see some noticeable discrepancy when the sample size is low (also confounding with sampling bias in weekly samples). Note that the discrepancy between mean and median trend estimates can be
mainly attributed to ozone heterogenous variability (Chang et al., 2023a), while the discrepancy between trend uncertainties can also be attributed to different optimized algorithms (if the regression assumptions are not severely violated, LS method tends to produce a narrower uncertainty than other algorithms, see Figure S4 for a further demonstration).

### 3.2   Strategies for improving trend detection: attribution of data variability

To further investigate the cause of sampling bias and improve the trend estimate, for the next step we aim to attribute the data
variability by incorporating meteorological variables and climate indices. Since meteorological variables are often correlated, we need to evaluate which variables have the best predictive performance, and determine a simple yet powerful model that accounts for the most variability.

   We investigate the impact of each climate index and meteorological variable on mean and median trends based on full sampling (Figure S5). Since the mechanism of incorporating covariates is highly similar between the mean and median regressions,
the following discussion is focused on mean trends only. Except for QBO, all the other variables show a trend over 1980-2021. Since a trend in the independent variable can induce a trend in the dependent variable, we also repeat the same analysis based on detrended covariates. After the trend from each covariate is removed, the trend estimates become more consistent (Figure S5). Overall, a stronger impact is found from dewpoint, relative humidity (in terms of much lower uncertainty), and ENSO (no improvement on signal-to-noise ratio, but it produces very different trends at shorter time periods 2000-2021 and 2005-2021);
and a weaker impact is found from other covariates. Quantitatively, dewpoint makes the greatest improvement by producing the lowest RMSD and MAD, and the highest signal-to-noise ratio for trend estimate (as previously shown in Chang et al. (2021)). By adding dewpoint alone, $R^2$ has increased from 0.54 (basic model) to 0.75 ($R^2$ for the full model is 0.77). This is



not unexpected because Chang et al. (2021) also showed that an incorporation of dewpoint produces a better fitting quality of ozone at MLO than relative humidity and temperature.

As discussed in Section 2.1, ozone variability at MLO is impacted by dry air masses from the north and west (low dewpoint), and moist air masses from the south and east (high dewpoint) (Gaudel et al., 2018), and its relative frequency is correlated with atmospheric circulations, such as ENSO (Lin et al., 2014). Therefore, in the following analysis, we select dewpoint and ENSO (in addition to the basic model M1) together as our best model for trend detection: Relative humidity and temperature are not included to avoid multicollinearity (these two variables jointly determine dewpoint); wind direction, wind speed and QBO are excluded because they have negligible impact on trend detection. We compare the model residuals from the basic and best models based on full sampling (Figure S6), and the result shows that the model fit is substantially improved after accounting for meteorology, in terms of (1) a reduction of RMSD and MAD by 27%, (2) the residual variability becomes weaker and the 2-sigma interval for Lowess (locally weighted scatterplot smoothing) curve becomes narrower, and (3) the nonlinearity in residuals is reduced. In addition, it is worth reinforcing that our sampling results are expected to be reliable in general, because the residuals are roughly linear over 1980-2021, which means there is no indication that the long-term trends have changed/turned around, and results are not sensitive to specific periods.

To differentiate from the basic trend model, we refer to the trend estimate from the best model as the meteorologically adjusted trend (since dewpoint is the main attributor). We then applied the best trend model to the once-per-week sampling test (see Figure 4 (b and d)). Through meteorological adjustments, this uncertainty attribution approach improves the precision of the ozone trends at all time scales, and the uncertainty is reduced by 35% on average (SD=7%) when focusing on full sampling. This approach also improves the accuracy of the trends which are based on once-per-week sampling, especially the great improvement of the trend bias from Tuesday sampling. These findings suggest that the sampling bias in trends can be substantially reduced through a consideration of colocated meteorological variability.

Additional analyses were carried out and described in the Supplementary Material. Specifically, (1) we show that an incorporation of colocated dewpoint observations produces a much better predictive performance than using monthly averages of all nighttime or 24-h data (Figure S7), demonstrating that some sampling bias can be attributed to meteorological variability; (2) temperature trends from each day of the week are more consistent with full sampling (Figure S8), emphasizing that more careful attention needs to be paid to ozone trend detection at low sampling frequency; and (3) an attribution analysis (Table S2) is carried out regarding pure sampling deviations (defined as the difference between an ozone daily value and its monthly mean). By comparing the magnitude of signal-to-noise ratio for each covariate, the result shows that a higher sampling variability is more likely to occur from July to November (with all covariates considered), but a weak $R^2$ of 0.39 indicates that a large portion of the sampling deviations might merely be unstructured variability.

## 3.3 Quantifying the benefit of increasing sampling frequency

This section extends the current scope beyond once-per-week sampling. The purpose is to investigate the strengths of different sampling schemes and eventually develop the best approach to reduce sampling bias with the minimal cost (i.e. fewer additional



samples). However, before this analysis is carried out, it is desirable to fully understand the relationship between enhancement of sampling rate and reduction of sampling bias.

We summarize the sampling strategies adopted in this study in Table 2. The most straightforward extension is to simply increase the sampling days per week (strategy A). The improvement on sampling bias in monthly means is provided in the
second part of Table 1. For the interpretation, (1) the 10% bias exceedance rate in monthly means is reduced from 29.4% to 11.9%, if we increase the sampling frequency from once-per-week to twice per week; (2) When focusing on the extreme cases, monthly mean bias exceedance above 25%, 20%, 15% and 10% can be eliminated based on 2-5 samples per week schemes, respectively; (3) 3 samples per week seems feasible in terms of limiting the 10% bias exceedance rate within 5% and limiting the average RMSPD and MAPD within 5% (or below 2 ppbv). In terms of sampling bias in trends for Strategy A, Figure 5
shows the mean trend and 2-$\sigma$ uncertainty for some cases based on 1-3 samples per week (another visualization is provided in Figure S9 by showing the full sampling variations for the periods 1990-2021 and 2000-2021, e.g., twice-per-week might also occur on Monday/Thursday, Monday/Friday, etc). From a sampling frequency of at least 3 days/week (or 43% data coverage), the trend estimate from each individual scenario is fairly close to the true trend (from full sampling), whereas quantitatively 5 samples per week are required to reduce the overall RMSPD and MAPD trend biases to 5% (see the first part of Table 3 and
later discussion).

In the next step we aim to quantify the marginal improvement of sampling bias by increasing the number of samples per month (Strategy B in Table 2). A complete random sampling is generally an infeasible plan in a monitoring program, but compared with the regular sampling Strategy A, the randomness in Strategy B enables us to avoid any potential systematic bias and investigate the undercoverage bias under different sampling rates. As discussed in Section 2.1, since the MLO data
set has limited missing values, it is reasonable to assume the resulting data characteristics and statistics are representative of the underlying population, i.e., not only monthly means, but also the trend estimate and its uncertainty derived from full sampling, should also be representative. Based on this rationale, it is possible to quantify the respective effect of sampling bias on (1) monthly mean, (2) trend estimate, and (3) trend uncertainty. The negative effect of sampling bias on trend estimate and uncertainty can be considered to be a deterioration of trend accuracy and precision, respectively. Since our focus is the marginal
improvement, the sampling frequency is the only control variable in this analysis (different time periods should play a minor role), so we only show the result based on the mean trends over 1990-2021 (it is sufficiently long such that the discrepancy between the mean and median trends is not critical).

Figure 6 shows the marginal decrement of bias exceedance rate and percentage bias as the monthly sampling rate increases (step curves), the colocated meteorological adjustments are made for all the trends and associated uncertainty estimates from
full and reduced sampling. This figure demonstrates that even if the weekly sampling frequency cannot be increased, any additional monthly samples can still reduce the sampling bias and improve the trend accuracy and precision, albeit the marginal improvement is generally leveling off at a certain point (as it is closer to the truth). Note that the step curves in Figure 6 are approximated results based on a random sampling over 10,000 iterations. Therefore, we use logistic regression to smooth out some heterogeneity and explicitly quantify the marginal improvement for each additional monthly sample. Let $p_x$ be
the bias exceedance rate or percentage bias derived from $x$ profiles-per-month, then logistic regression can be expressed as



$\log\left(p_x/(1-p_x)\right) = a + bx$, where $a$ and $b$ are coefficients to be fitted. Since the coefficient $b$ in logistic regression cannot be directly interpreted as a slope or marginal effect, an average derivative is used to represent the marginal effect (by calculating a derivative of the curve for each $x$ and then taking the average, Kleiber and Zeileis (2008)). An alternative approach is to use odds ratio ($e^b$), but odds ratio is often misinterpreted as probability, and it is not a standard measure in the atmospheric science

literature, so odd ratio is not adopted here. In summary, the marginal improvement is roughly consistent between monthly mean, trend estimate and uncertainty, each additional monthly sample corresponds to an average reduction of bias exceedance rate by 3.3% (SD=1.6%) and percentage bias by 0.6% (SD=0.2%).

  Note that an offset can be observed for the 5% bias exceedance rate in Figure 6(b), due to variability in the colocated dewpoint observations (standard regression model assumes that covariates are measured without error), which can cause some

weak inconsistency in trend estimates (if we use the basic trend model, such an offset can be removed, see Figure S10). Nevertheless, since the other metrics are not affected, there is no need to particularly adjust the 5% bias exceedance rate in this analysis. If necessary, errors-in-variables models can be applied (Gleser, 1981; Li, 2002).

  The exercise in Figure 6 can be used as a reference to determine the minimum number of samples necessary for accurate and precise trend detection. For example (based on logistic regression fits), at least 12 (and 23) samples per month are required

to reduce MAPD trend bias to 10% (and 5%); and at least 15 (and 26) samples are required if RMSPD is a criterion. Based on Figure 6 and compared with previous studies, (1) we found that a roughly 10% (and 5%) sampling uncertainty in monthly mean (MAPD) is associated with a sampling frequency of 4 (and 12) samples per month, which is consistent with the IAGOS profiles at 700 hPa above Frankfurt, Germany (Saunois et al., 2012); (2) 25 samples per month are necessary for producing the mean value within ±2% MAPD bias, which is consistent with the regional study above western North America (Cooper

et al., 2010); (3) Chang et al. (2020) found that 18 samples per month are required for the MAPD trend bias to be less than 5% above Europe; under the same frequency the MAPD trend bias is 6.8% at MLO, and it takes an additional 5 monthly samples to reach the 5% threshold, as the marginal improvement eventually becomes less effective. Therefore, instead of determining the minimum frequency necessary for trend detection purely based on a predetermined threshold, it would be preferable to use an efficient sampling strategy to objectively determine the minimum data coverage (see next section).

**3.4 Cost-benefit strategies for effectively increasing sampling rate**

For the situation when 3 regular samples per week or higher are too expensive to maintain for a given ozonesonde station, we aim to develop a simple cost-benefit strategy to improve upon the existing once-per-week sampling scheme, by efficiently reducing sampling bias in monthly means and trends with the minimal cost. Two strategies are proposed in order to achieve this goal (see Table 2). Strategy C is designed for increasing additional samples during a predetermined season (e.g., ozone is

typically more variable and less predictable during seasons with frequent stratosphere-troposphere exchange events). Strategy D adopts an anomaly detection approach to temporarily increase sampling rate based on the sampling deviation against the climatology. The rationale is to first develop a baseline climatology (i.e., monthly mean and SD over 1980-1989). Then for the period 1990-2021 if any new weekly sample (the first in a week) is too extreme compared to the climatology (e.g., outside monthly mean ± 2SD), we take a second sample two days after the initial sampling date (a third sample can be taken two days



after the second sampling date if necessary); otherwise no extra sample within a week is required. Hereafter we only show the result based on mean trends over 1990-2021 when comparing different sampling strategies.

Strategy C is a mixed sampling approach in which we use once-per-week sampling for all months as the baseline, then during a particular season the frequency is increased to 2-7 samples per week (while the other seasons maintain once-per-week sampling). In Figure 7 we show the seasonal trends over different time periods and seasonal trend bias over 1990-2021 (difference between subsampled trend and the true trend, with meteorological adjustments). This is unexpected that no consistent improvement on trend precision and accuracy can be seen from extra seasonal samples. Even if we increase the seasonal samples up to full sampling, improvement can only be achieved in Jun-Jul-Aug (JJA) and Dec-Jan-Feb (DJF), while Mar-Apr-May (MAM) and Sep-Oct-Nov (SON) still show a strong bias (and no better than once-per-week sampling). Table 3 shows the percentage bias from Strategies A and C (5 samples per week for a particular season and once-per-week for other seasons, so the coverage rate is similar to twice-per-week for all months): From this table we can see that even though monthly mean bias is reduced from Strategy C (compared to once-per-week sampling), trend bias is unexpectedly increased in MAM and SON.

The most likely reason for the trend bias in MAM and SON is that these two seasons represent the MLO ozone peak and trough of a year, so if we only enhance the sampling in either season, the seasonal variability tends to outweigh the trends (analogue to only sampling the tail of a histogram). In contrast, seasonal variability in JJA and DJF is more consistent with the overall mean, thus the trend estimates are more likely to be improved (see Figures S11-S13 for detailed demonstrations). The cause of such phenomena can be associated with a well-known sampling fallacy, also known as selection bias (Bateson and Schwartz, 2001) or preferential sampling (Diggle et al., 2010). This fallacy typically occurs when the samples are heavily biased toward a specific subset of the target population. Since there is no guarantee on the improvement of trend bias, Strategy C is not an ideal sampling approach.

Although we found no direct causality between a reduction of monthly mean bias and trend bias, it is natural to suspect that the extreme sampling biases attached to certain months might be the main attributor to trend bias. Strategy D is an adaptive approach designed for aiming at elimination of sampling bias that is too great to be reasonable or acceptable. As discussed above, we use 1980-1989 ozone data to develop a baseline climatology (i.e. seasonal cycle and SD) and 1990-2021 data to validate the result. Given the fact that the seasonal cycle is accounted for, a tolerance range needs to be defined to determine if the magnitude of the desesasonalized anomaly is reasonable. The narrower the tolerance range (the higher the sampling rate), the greater reduction the extreme sampling bias (see Figure S14). With this strategy, if we assume at most 3 samples per week, our aim is to find an optimal tolerance range, such that the trend bias is better than the 3 regular samples per week scheme (indicating that fewer samples are used to achieve a better estimation). Since the climatological mean and SD vary at different months, the sampling rate is not uniformly distributed across the year. Figure 8 displays the climatology and average monthly sample size based on different tolerance ranges and at most 3 samples per week: We can see that the necessary monthly samples are clearly associated with the monthly variability: the higher the monthly SD (e.g., April), the less additional samples are required. Our specification for tolerance is purely based on ozone monthly variability, alternative approaches might be





possible if the extreme sampling bias can be attributable to other factors (e.g. extreme weather conditions, but it is beyond the scope of this study).

To facilitate discussion of Strategy D, let X(+Y):Z$\sigma$ denote a sampling scheme based on X regular samples per week with at most Y extra samples per week according to Z-$\sigma$ tolerance range (a demonstration is made by the 1(+2):0.5$\sigma$ scheme in Figure 9). The percentage bias and coverage rate for Strategy D are summarized in the third part of Table 3:

– When focusing on at most 2 samples per week, the trend accuracy based on the 0.5-$\sigma$ tolerance ($\sim$1.8/7) is comparable to 2/7 regular sampling.

– When focusing on at most 3 samples per week, the trend accuracy based on the 1-$\sigma$ tolerance ($\sim$2.2/7) is comparable to 3/7 regular sampling. The trend bias can be further reduced to below 5% if a narrower 0.5-$\sigma$ tolerance ($\sim$2.5/7) is applied (5/7 regular sampling is required to meet the same goal).

Under these circumstances, an improved trend accuracy is achieved with fewer samples than regular sampling, but the bias metrics for monthly means and trend uncertainty remain at similar levels as regular 2/7 or 3/7 sampling; this result indicates that Strategy D is only designed for improving trend accuracy. Note that from the above result a more constrained tolerance is required to achieve our goal, but a gradual improvement can still be provided by the 2- or 1.5-$\sigma$ tolerance. This suggests that the adaptive sampling strategy can be tailored to a specific sampling rate according to the budget (by modifying the tolerance range and the maximal samples allowed in a week, see next section).

### 3.5 Recommendations on efficient sampling for trend detection

We recognize that, in reality, once-per-week sampling does not imply that the sample is always measured on the same day of the week. We use the ozonesondes launched at Hilo, Hawaii, as an example (19.72°N and 155.05°W, with a roughly once-per-week sampling frequency in 1982-2021): The effect of (real) sparse sampling is shown by matching the Hilo ozonesonde launch dates and the MLO surface ozone record, then the MLO ozone trends are estimated based on the Hilo ozonesonde sampling dates and also by shifting 1, 2,.., 6 days after the colocated dates (see Figure S15). The result shows that a strong sampling bias on trends can be observed from the Hilo ozonesonde sampling scheme, and also based on our previous finding that similar amounts of sampling variability can be observed between the Hilo ozonesonde and MLO nighttime ozone records (Figure S1), therefore the implications for undercoverage bias from previous discussions are still valid.

We revisit the analysis in Figure 5 by incorporating the anomaly-based sampling strategy. Figure 10 shows the trends based on the 1(+1):0.5$\sigma$, 1(+1):1$\sigma$ and 1(+2):1$\sigma$ schemes, respectively. This demonstrates how we can tailor our sampling scheme to a specific budget by adjusting the tolerance range and the maximal samples per week (additional demonstrations are provided in Figure S16). These selections are made because previous comparisons (based on 1990-2021) found that the trend accuracy is comparable between 2 regular samples per week and 1(+1):0.5$\sigma$ scheme, and between 3 regular samples per week and 1(+2):1$\sigma$ scheme. So we might be able to reduce the cost without compromising the trend accuracy by omitting some samples which are expected to convey less new information.



In summary, to achieve efficient sampling enhancement for trend detection based upon current once-per-week sampling, we can adjust our adaptive sampling strategy according to the funds available to purchase additional ozonesondes: In terms of annual samples and budget (the cost to launch one ozonesonde is roughly \$1500, to cover both equipment and personnel costs), we can make the best use of an extra 26 profiles in a year (~1.5/week) by applying the 1(+1):2$\sigma$ scheme. If an additional 42 profiles per year (~1.8/week) are allowed, we can achieve a similar trend accuracy as to 2 regular samples per week by applying the 1(+1):0.5$\sigma$ scheme. Likewise, an additional 63 profiles (~2.2/week) allow us to produce a trend accuracy comparable to 3 regular samples per week (the 1(+2):1$\sigma$ scheme) (3 profiles per week is equal to 156 profiles per year). Therefore, our ultimate goal can be achieved without fully sampling 3 days per week (and saving the cost for 41 profiles a year). Although our recommendations are based on the MLO ozone data variability, an adjustment tailored to a specific time series can be made as long as the climatology can be reliably determined. Previous work has shown that models with couples stratosphere-troposphere dynamics and chemistry can realistically simulate ozone variability in the free troposphere for the purposes of evaluating sampling strategies and the impact of interannual variability on long-term ozone trends (Lin et al., 2015; Barnes et al., 2016). Since our baseline reference constitutes both climatological mean and SD, it can be adapted to the environments at different locations (as the climatology from each monitoring site is expected to reflect local features and variations).

There is no additional statistical complexity to extend the anomaly-based sampling strategy to vertical profile data, except that we translate the comparisons between surface measurements into vertical profiles (either MAD or RMSD can be used to represent the average deviation between individual profiles and the climatology at a fixed vertical grid). Nevertheless, one significant complication in profile analysis is the common presence of stratospheric intrusions, as these events can greatly enhance the ozone concentrations in the mid- and upper troposphere, and occur more frequently in spring and early summer (see Figures 1 and S1). Data measured during those events are highly leveraged and usually filtered out for tropospheric ozone trend analysis (albeit these data are valuable for studying stratosphere-troposphere exchange processes). Therefore additional samples are required in regions frequented by stratospheric intrusions, because these data are more likely deviant from the climatological ranges. Therefore, the optimal sampling strategy will be affected by the frequency of stratospheric intrusions.

## 4 Conclusions

This paper used over 40 years of daily nighttime ozone data measured at Mauna Loa Observatory (representative of the lower free troposphere) to show that a large trend bias could be present when the sampling frequency is sparse and insufficient. Although the variability of sampling deviations is rather unpredictable over time (dependent on meteorology to a certain extent), these sampling deviations become inherent biases in a time series when samples are limited. Since the trend estimate is derived by the chronological order of a time series of events, certain extreme sampling biases attached to different times could have a severe impact on trend estimation. In this study we have shown two remedies to improve trend detection under sampling bias: the first is to attribute the data variability by incorporating colocated meteorological variables, demonstrating a substantial improvement in trend precision (and a moderate improvement for the trend accuracy); and the second is to adopt an adaptive sampling strategy for eliminating anomalous sampling bias, which allows improved trend accuracy with fewer samples.



We summarized the challenges of detecting free tropospheric ozone trends as follows:

1. At least a few decades of continuous data could be necessary to confidently detect a weak signal of ozone trends. However, if sampling bias is present, an extra period of time might be required to be able to detect the same signal. In our first trend result, we used the MLO ozone data to show that highly confident ozone trends (2-sigma confidence level) can be detected over 1990-2021 under full sampling, but under once-per-week sampling, trends might fail to be detected even though an additional 10 years of data is considered (the left panels of Figure 4, for both mean and median estimators).

2. The longer the time period, the more consistent the trend estimates from different trend techniques (mean and median regression), but sparse sampling can result in similar biases to long-term trends regardless of trend techniques.

3. A proper attribution of data variability can efficiently improve the trend precision. Based on full sampling ozone data at MLO, meteorological adjustments improve residual RMSD and MAD by 27%, and reduce the trend uncertainty by 35% (an average over different time periods). In terms of variable selection, dewpoint is the most important variable for ozone trend detection and attribution at MLO, as it is a good indicator of air mass origin and it removes the noise in the data caused by the constant shifting between air masses of mid-latitude or tropical origin (Gaudel et al., 2018).

4. Meteorological adjustments can also reduce the undercoverage bias due to sparse sampling, and thus improve the trend accuracy. This conclusion is drawn because once-per-week sampling trend bias is largely reduced after meteorology is accounted for (the right panels of Figure 4), and the trends are more consistent between different days of the week.

5. An incorporation of climate indices, such as ENSO and QBO, has no effects on the improvement of sampling bias, because these large scale circulations are only characterized at monthly level. In contrast, we showed that a better predictive performance can be achieved by incorporating the colocated dewpoint observations (at the same sampling dates), compared to using monthly aggregated information. This result indicates that small-scale colocated adjustments are more important for reducing undercoverage bias.

6. When a regular sampling strategy is adopted, we found that 3 samples per week is required to (1) reduce 10% exceedance bias and the overall monthly mean bias to below 5%, and (2) avoid extreme bias in trends (compared to 1 or 2 samples per week), while 5 samples per week are required to reduce the trend bias (both RMSPD and MAPD) to below 5%.

7. Imbalance sampling might deteriorate the trend accuracy due to selection bias. We used once-per-week sampling as a reference, and increased additional regular samples at a particular season; The result shows that an improved trend estimate can only be achieved in JJA and DJF, while the trend bias is deteriorated in MAM and SON.

8. We proposed an adaptive sampling approach that adopts an enhanced sampling frequency if any upcoming sample is too deviant from the baseline climatology. By eliminating extreme sampling bias, this approach can efficiently improve the trend accuracy with fewer samples (an average 2.2 samples per week) than a regular sampling strategy of 3 samples per



week. If we use a more constrained tolerance (an average of 2.5 samples per week) to rule out the extreme sampling bias, the RMSPD and MAPD trend bias can be reduced to 5%.

It should be emphasized that the effect of undercoverage bias summarized above is to be expected in a sparsely sampled environment even if perfect observations are obtained (i.e. no measurement uncertainty). The general implications are expected
to be the same for vertical profile trend analysis, since consistent amounts of sampling variability are observed from intensive sampling campaigns (Figure 1) and decadal seasonal variability (Figure S1).

Looking to the future, the sampling strategy proposed in this study is designed to validate the trace gas products produced from NOAA's current Joint Polar Satellite System (JPSS, https://www.nesdis.noaa.gov/our-satellites/currently-flying/joint-polar-satellite-system), future Geostationary Extended Observations (GeoXO satellite system, scheduled for launch in
the early 2030s, https://www.nesdis.noaa.gov/GeoXO), and the future Near Earth Orbit Network (NEON, scheduled for launch in the early 2040s). Previous efforts to compare trends between ground-based and satellite measurements are typically based on aggregated monthly time series (Gaudel et al., 2018). This study shows that such a comparison could be biased when the sampling rate is low or the sampling schemes are different. Given the current free tropospheric ozone observing system is sparse not only in time, but also in space (Tarasick et al., 2019), it is questionable whether the existing network is capable of
comprehensively performing satellite evaluation and validation. Therefore, in addition to a proper sampling rate, a reliable and extensive monitoring network is also required (Weatherhead et al., 2018).

Previous designs to expand monitoring network or spatial coverage were typically determined through observation correlation ranges (Sofen et al., 2016; Weatherhead et al., 2017). This type of analysis aims to minimize the spatial gap by maximizing correlation ranges from additional sites, but these additions are not designed to accurately evaluate global or regional trends.
Specifically, we point out that strong spatial heterogeneity is often present in regional ozone trends and variability, as indicated by a wide range of free tropospheric trends observed at individual sites above Europe and western North America (Chang et al., 2022, 2023a), and therefore evaluating regional ozone trends based on a single sparsely-sampled data source is likely to produce an incomplete assessment of the true trend. We thus recognize the importance of evidence synthesis by integrating data from various platforms (Richardson, 2022; Shi et al., 2023). For instance, aircraft field campaigns are mostly short-term or tem-
porary activities, but those data are carefully planned with specific science objectives (such as improving forecasting skill and evaluating satellite data), thus those data should also be considered in the regional trend assessment, together with ozonesonde, lidar and commercial aircraft data sets (through a detailed data intercomparison and data fusion approaches (Cooper et al., 2010; Liu et al., 2013; Chang et al., 2022, 2023a)).

*Code and data availability.* MLO meteorological data can be found at ftp://aftp.cmdl.noaa.gov/data/meteorology/in-situ/mlo/ (NOAA GML,
2023a), and ozone data can be found at https://gml.noaa.gov/aftp/data/ozwv/SurfaceOzone/ (NOAA GML, 2023b). Ozonesonde data measured at Trinidad Head (California) and Hilo (Hawaii) can be downloaded at ftp://aftp.cmdl.noaa.gov/data/ozwv/Ozonesonde/ (NOAA GML, 2023c). The Python and R codes for implementing quantile regression based on the MBB algorithm are provided in the TOAR statistical guidelines (Chang et al., 2023b).





*Author contributions.* KLC conducted the analysis. KLC and ORC contributed to the conception/design and drafted the paper, while GM and BCM helped with the revision. AG, IP and PE contributed to the acquisition of data. All authors approved the submitted and revised versions for publication.

*Competing interests.* ORC is the Scientific Coordinator of the TOAR-II Community Special Issue, to which this paper has been submitted, but he is not involved with the anonymous peer-review process of this or any of the other papers submitted to the Special Issue journals.

*Acknowledgements.* KLC, ORC, AG, IP and PE are supported by NOAA cooperative agreement NA22OAR4320151. We acknowledge support from the NOAA JPSS PGRR program.



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





**Table 1.** Undercoverage bias in monthly means (1980-2021), based on (a) once-per-week sampling by each day of the week and (b) different sampling frequencies per week (average over all possible subsets). Bias exceedance rate indicates the frequency that the absolute sampling bias of the mean from reduced samples is greater than a threshold (5% - 25%) of the mean from complete samples.

| | Bias exceedance rate [%] | | | | | RMSPD | MAPD | RMSD | MAD |
|---|---|---|---|---|---|---|---|---|---|
| (a) Sampling day | 5% | 10% | 15% | 20% | 25% | [%] | [%] | [ppbv] | [ppbv] |
| Sun | 59.7 | 29.2 | 13.3 | 5.2 | 2.6 | 10.2 | 8.0 | 4.2 | 3.3 |
| Mon | 55.2 | 27.8 | 11.5 | 5.0 | 1.8 | 10.1 | 7.7 | 4.1 | 3.2 |
| Tue | 59.7 | 31.5 | 15.5 | 5.8 | 1.4 | 10.6 | 8.3 | 4.3 | 3.4 |
| Wed | 59.1 | 31.7 | 16.1 | 6.0 | 3.0 | 10.6 | 8.3 | 4.4 | 3.4 |
| Thu | 55.2 | 26.6 | 10.9 | 3.8 | 1.0 | 9.4 | 7.3 | 3.9 | 3.1 |
| Fri | 61.1 | 29.2 | 13.7 | 5.2 | 2.6 | 10.5 | 8.2 | 4.4 | 3.4 |
| Sat | 60.3 | 30.0 | 13.9 | 4.8 | 2.4 | 10.4 | 8.2 | 4.3 | 3.4 |
| (b) Sampling frequency | | | | | | | | | |
| One day per week | 58.6 | 29.4 | 13.5 | 5.1 | 2.1 | 10.3 | 8.0 | 4.2 | 3.3 |
| Two days per week | 39.4 | 11.9 | 3.2 | 0.7 | 0.3 | 6.6 | 5.1 | 2.7 | 2.1 |
| Three days per week | 25.6 | 4.4 | 0.7 | 0.1 | 0 | 4.8 | 3.7 | 2.0 | 1.5 |
| Four days per week | 14.6 | 1.3 | 0.1 | 0 | 0 | 3.6 | 2.7 | 1.5 | 1.1 |
| Five days per week | 6.0 | 0.2 | 0 | 0 | 0 | 2.6 | 2.0 | 1.1 | 0.8 |
| Six days per week | 0.7 | 0 | 0 | 0 | 0 | 1.7 | 1.3 | 0.7 | 0.6 |



**Table 2.** Sampling strategies adopted in this study.

| Strategy | Description |
| --- | --- |
| A | *A fixed sampling frequency within a week*: |
| | A weekly sampling frequency ($d = 1, \cdots, 7$) is predetermined to conduct the sampling analysis. For example, if $d = 2$ and Monday/Wednesday are chosen, then we select the data measured in corresponding days of the week to produce the monthly means and trends. |
| | Since the possible combinations for this scheme are small (i.e., 7 possibilities for choosing 1 day/week and 6 days/week, 21 possibilities for choosing 2 days/week and 5 days/week, and 35 possibilities for choosing 3 days/week and 4 days/week), the sampling results are based on all possibilities. |
| B | *A fixed sampling frequency within a month*: |
| | A monthly sampling frequency ($d = 2, \cdots, 29$) is predetermined to conduct the sampling analysis. For example, if $d = 15$, we randomly choose 15 different days of the month, then we select the data measured in corresponding days of the month to produce the monthly means and trends. |
| | Since the possible combinations for this scheme can be very large (there are over $1.5 \times 10^8$ combinations for choosing a subset of 15 out of 30 days), the sampling results are based on 10,000 times of random resampling. |
| C | *A seasonally sampling enhancement*: |
| | Based on existing once-per-week sampling, an increased sampling is applied to a particular season (e.g., twice-per-week sampling in March-April-May and once-per-week sampling for other seasons). |
| D | *An adaptive sampling strategy according to the deviation from the climatology*: |
| | The procedure can be outlined as follows: |
| | – A baseline monthly climatology needs to be established (i.e., see Figure 8 as a demonstration). |
| | – Based on existing once-per-week sampling, if a new weekly sample is too extreme to be acceptable (e.g. outside a threshold from the climatology), then we take an additional sample two days later; otherwise no further sample within a week is required. |
| | – By adjusting the threshold and the maximal affordable samples per week, we aim to efficiently reduce sampling bias in monthly means and trends with the minimal additional samples. |





**Table 3.** Undercoverage bias in monthly means, trend estimate and trend uncertainty from different sampling strategies (1990-2021, with meteorological adjustments): Strategy A is based on different sampling days per week for all months. Strategy C is based on once-per-week sampling for all months, incorporated with additional 4 days-per-week sampling in a specific season (so the data coverage is similar to two days/week sampling for all months). Strategy D is an anomaly based strategy: X(+Y):Z$\sigma$ denote a sampling scheme based on X regular samples per week with at most Y extra samples per week according to Z-$\sigma$ tolerance range.

| | | monthly mean bias | | trend bias | | trend uncertainty bias | |
|---|---|---|---|---|---|---|---|
| Strategy A | Coverage [%] | RMSPD | MAPD | RMSPD | MAPD | RMSPD | MAPD |
| One day per week | 14.3 | 10.4 | 8.1 | 13.4 | 11.1 | 32.1 | 30.6 |
| Two days per week | 28.5 | 6.6 | 5.1 | *8.7* | *7.3* | 16.9 | 15.5 |
| Three days per week | 42.7 | 4.8 | 3.7 | **6.6** | **5.4** | 10.4 | 9.3 |
| Four days per week | 57.0 | 3.6 | 2.8 | 5.6 | 4.6 | 7.1 | 6.0 |
| Five days per week | 71.2 | 2.6 | 2.0 | 4.6 | 3.8 | 6.2 | 5.7 |
| Six days per week | 85.4 | 1.7 | 1.3 | 4.1 | 3.6 | 4.2 | 3.5 |
| Strategy C | | | | | | | |
| MAM(5)+others(1) | 28.7 | 9.4 | 6.8 | 16.8 | 14.6 | 23.8 | 21.5 |
| JJA(5)+others(1) | 28.7 | 8.4 | 6.2 | 7.6 | 6.3 | 24.0 | 22.1 |
| SON(5)+others(1) | 28.5 | 9.0 | 6.5 | 20.2 | 18.0 | 25.7 | 24.2 |
| DJF(5)+others(1) | 28.4 | 9.7 | 7.0 | 8.9 | 6.9 | 28.3 | 27.0 |
| Strategy D | | | | | | | |
| 1(+1):2$\sigma$ | 20.8 | 8.5 | 6.5 | 13.4 | 10.9 | 24.6 | 23.4 |
| 1(+1):1.5$\sigma$ | 22.4 | 7.9 | 6.1 | 12.7 | 10.5 | 23.1 | 21.7 |
| 1(+1):1$\sigma$ | 24.1 | 7.4 | 5.7 | 10.9 | 8.9 | 20.6 | 18.9 |
| 1(+1):0.5$\sigma$ | 25.9 | 6.9 | 5.3 | *7.0* | *5.5* | 16.8 | 16.1 |
| 1(+2):2$\sigma$ | 24.3 | 7.9 | 6.1 | 10.4 | 7.5 | 21.4 | 19.9 |
| 1(+2):1.5$\sigma$ | 27.5 | 7.1 | 5.4 | 8.8 | 6.3 | 18.2 | 16.4 |
| 1(+2):1$\sigma$ | 31.3 | 6.2 | 4.7 | **6.4** | **5.0** | 16.7 | 15.3 |
| 1(+2):0.5$\sigma$ | 35.9 | 5.2 | 3.9 | **3.6** | **2.7** | 14.7 | 12.1 |



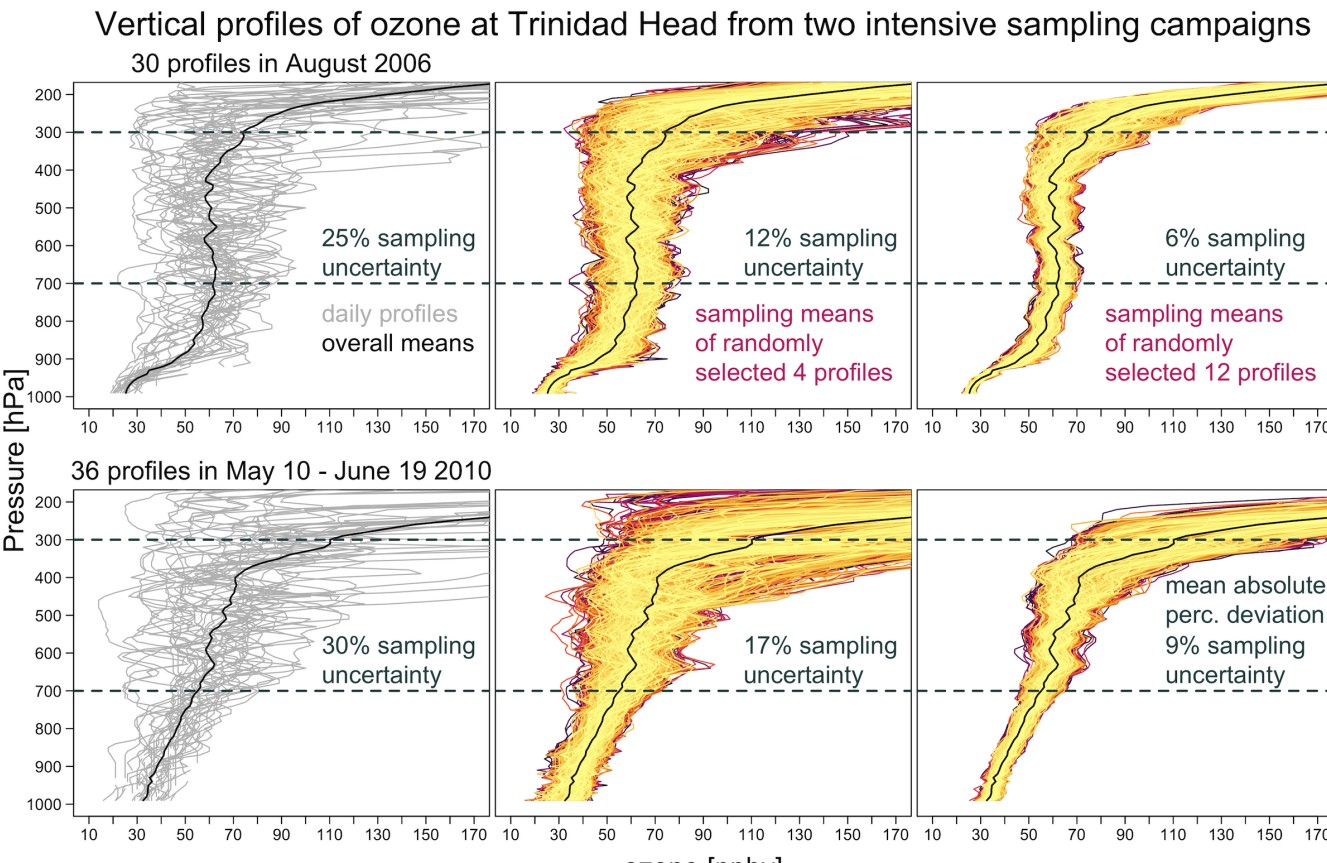

**Figure 1.** Demonstration of ozone variability from two intensive sampling campaigns (30 profiles in August 2006 and 36 profiles in May 10 - June 19 2010 at Trinidad Head, California) and sampling variability of subsampled means: Individual sondes and the overall means are shown in the left panels; variabilities of subsampled means are generated in the middle and right panels by randomly selecting 4 or 12 sondes over 1000 times, respectively. This analysis demonstrates that the sampling uncertainty on monthly means in the free troposphere can be reduced by half if the samples are increased from 4 to 12 sondes a month (evaluated by mean absolute percentage deviation at 10 hPa resolution layers).





**Figure 2.** Demonstration of sampling bias from once-per-week sampling on monthly means and trends over 1980-2021 (nighttime temperature and ozone at MLO). Each point represents a monthly mean, aggregated from full sampling (black), or once-per-week sampling taken just on Sunday (red) or Tuesday (blue). Each vertical range represents the magnitude of sampling bias in a given month. Trends and associated uncertainty estimates are based on the basic model (M1).



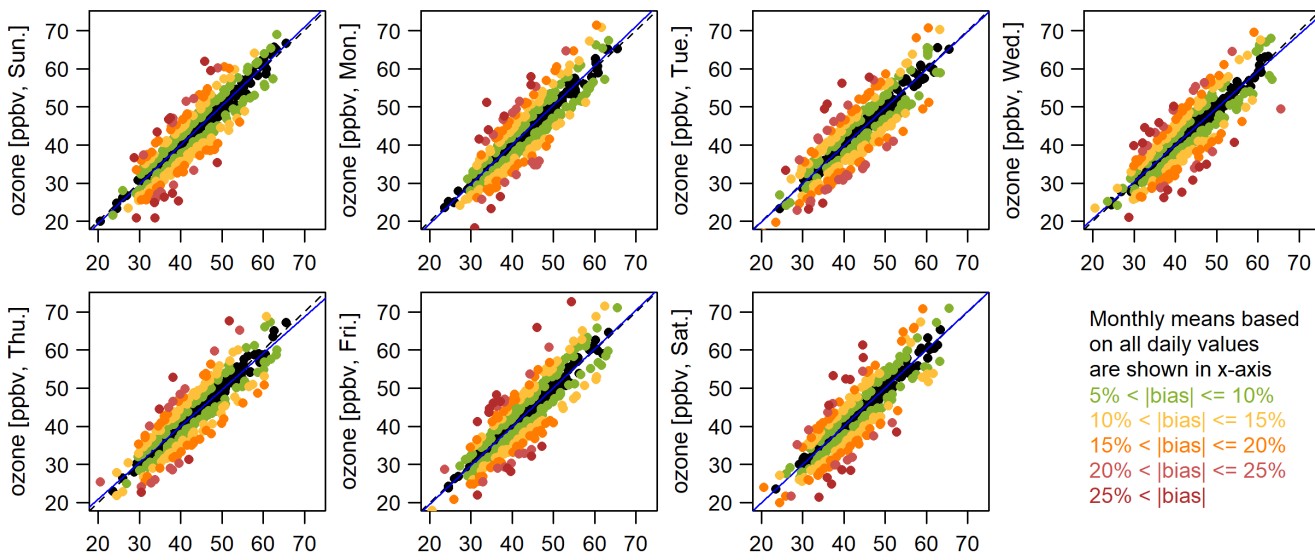

**Figure 3.** Demonstration of exceedance bias (at different thresholds) from once-per-week samples on monthly means: Panels show the scatter plots between monthly means based on full sampling (x-axis) and once-per-week sampling (y-axis, by each day of the week). Result is based on daily nighttime observations measured at MLO (1980-2021). Blue line in each panel represents the overall correlation.





**Figure 4.** MLO ozone trends and 2-sigma intervals derived from the mean (a & b) and median (c & d) estimators, without (a & c) and with (b & d) meteorological adjustments, respectively. In each panel the results are based on monthly means aggregated from full sampling (labeled as 'all') and once-per-week sampling (labeled by day of the week) for six different time periods.



**Figure 5.** Same as Figure 4, but based on the mean estimators only and (from top to bottom) 1-3 samples per week (e.g., a label '1,3,5' indicates sampling on Sunday, Tuesday and Thursday), respectively.





**Figure 6.** Marginal decrement of the bias exceedance rate (upper panel), and RMSPD and MAPD (lower panel) in monthly means, trend estimates, and trend uncertainties, according to different sampling frequencies per month (the MLO nighttime ozone record, 1990-2021): Step curves represent the results obtained from resampling method, and smooth curves represent the logistic regression model fit in order to quantify the marginal improvement. Trends and associated uncertainty estimates are meteorologically adjusted.



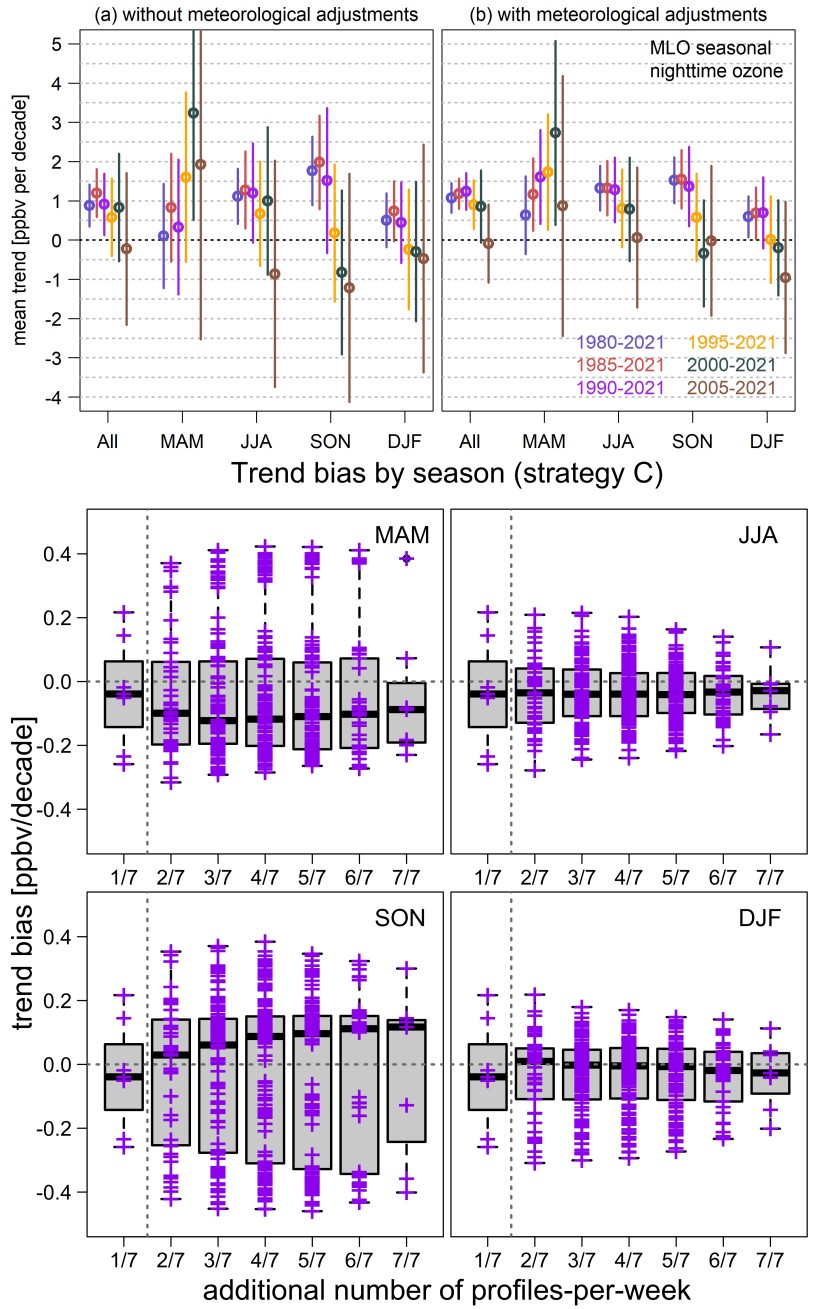

**Figure 7.** MLO ozone seasonal trends under full sampling (upper panel) and seasonal trend bias (each cross represents a difference between subsampled trend and the true trend, with meteorological adjustments, lower panel) under mixed sampling (1990-2021, Strategy C): 1/7 indicates the baseline scenario representing once-per-week sampling for all months, and $k/7 (k = 2, \cdots, 7)$ indicates $x$ samples per week for a particular season, while the other seasons remain once-per-week sampling.





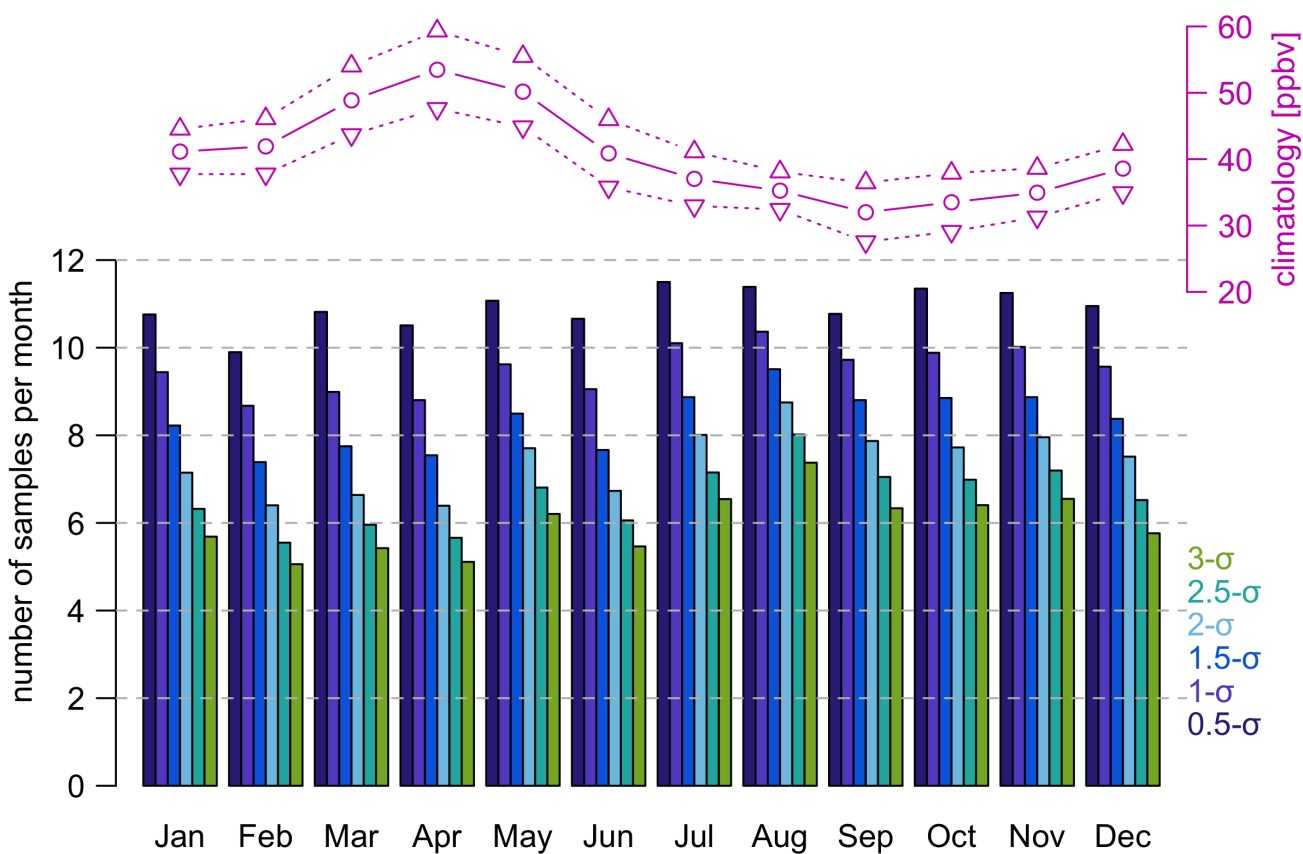

**Figure 8.** Monthly climatology [±1 SD] and average samples per month based on at most 3 samples per week and different tolerance ranges (Strategy D).







**Figure 9.** Demonstration of anomaly-based sampling for ozone time series: The upper panel shows the magnitude of monthly sampling bias between full sampling (black) and once-per-week sampling on Sunday (purple). The lower panel is based on the same scheme, but additional samples are taken when any weekly samples are found to be outside the 0.5-$\sigma$ range from the climatology (at most 3 samples per week). Trends and associated uncertainty estimates are meteorologically adjusted.





**Figure 10.** Same as Figure 5, but anomaly-based sampling strategy is incorporated: from top to bottom shows the 1(+1):0.5σ, 1(+1):1σ and 1(+2):1σ schemes, respectively. Numbers in parentheses in the x-axis indicate the extra sampling days of the week.