# Peer review of "Technical note: Challenges of detecting free tropospheric ozone trends in a sparsely sampled environment"

_EGUsphere, 2023_

## Author Response (AR1)

We thank the referees for the obvious time and care they put into their reviews, which helped us to revise the manuscript with improved focus and clarity. We have addressed all of the referee comments as described below. The reviewer comments are shown in bold font, followed by our response in normal font.

**Anonymous Referee #1**
**This is an interesting study showing how sparse sampling may affect trends and monthly means of ozone in the troposphere. We all have to live with the limitations of our observing system. This study explicitly shows some of them, and also gives some ideas about possible improvements, by using additional parameters, or additional observations at suitable times. All of this is scientifically interesting and I commend the authors for a well written and well illustrated paper. Unfortunately, we also know that many observing systems are controlled by factors much different from the statistical properties of the observed atmospheric property, e.g. ozone. So while the study is very good and interesting, I wonder how much change, e.g. in ozone sounding frequency it could, or should, generate. Hopefully not a reduction.**

**Apart from this more general question, I don't really have any detailed comments. I think the authors have done a very good job, probably with a few iterations.**

We thank the reviewer for the positive feedback. In this study we aim to do our best to convince the readers that reliable ozone trend assessment should be based on an adequate sampling rate (in most cases this is not met), and that increasing the sampling frequency is as important as maintaining continuous records and expanding observational networks. To better address this comment, we have replaced Figure 6 in the submitted version with a more elucidated demonstration (it is the same analysis, but is shown in a different way, i.e., based on fundamental statistical metrics, such as statistical power/type II error; and to avoid introducing an additional concept, such as logistic regression). We consider this to be a powerful overview of the challenges associated with a low sampling rate, and clearly emphasizes that a regular sampling frequency of 3 samples/week is a desirable (minimal) requirement.

The discussion associated with the new Figure 6 is added as follows (the obsolete version of Figure 6 along with the relevant discussion is relegated to the supplemental material):

[revised manuscript text omitted]

Saunois et al.: Impact of sampling frequency in the analysis of tropospheric ozone observations, Atmos. Chem. Phys., 12, 6757–6773, https://doi.org/10.5194/acp-12-6757-2012, 2012.

Weatherhead et al..: Factors affecting the detection of trends: Statistical considerations and applications to environmental data, J. Geophys. Res.-Atmos., 103, 1998.

**Anonymous Referee #2**
**This paper is a comprehensive sensitivity study of the impact of sampling rate on the accuracy and precision of inferred long-term trends. The paper's syntax and structure are clear, although from time to time, the excessive amount of details makes it more difficult to follow.**

**The authors start with a high-elevation, multi-decadal surface ozone timeseries (Mauna Loa Observatory, 19.5N) known to have a high sampling frequency with very few gaps over several decades (the "perfect timeseries"), then design a number of scenarios in which only a fraction of all samples are used to compute the trends using classic multi-component LS and LAD fitting models (collocated dew point found to be the most critical proxy after deseasonalizing). The monthly mean bias, and trend bias and error are computed for all scenarios to assess which sub-sampling scenarios yield best agreement with the full-sampling results. The scientific approach is excellent and the authors try to address a very important and well-known aspect of atmospheric composition trend in general, which means that their findings are potentially applicable to a lot more studies beyond free tropospheric ozone.**

We thank the reviewer for the positive feedback.

**If there must be one main criticism, it is only to say that the results are extremely detailed, too detailed at times, to a point that many figures could be simplified in order for the reader to extract the essential information. For example, I suggest that the "without meteorological adjustment" figures be removed after section 3.2. Another example, is the number of time periods, ranging from 1980-2021 to 2005-2021. I think showing results for only 2 or maximum 3 periods is just enough and will avoid overloading many figures.**

To address the issue of overloaded details, we have made the following revisions:
1. Since the variable selection process in Section 3.2 can be independent from our discussion of sampling bias (but still considered to be important), the relevant discussion is moved to Appendix A.
2. The discussion in Section 3.3 is rewritten and substantially reduced (with much less technical details, also see the response to reviewer 1). This allows us to avoid introducing an additional statistical concept (e.g. logistic regression), and instead place the focus on sampling impact.
3. Discussion is mainly focused on meteorologically adjusted trends in Sections 3.3 - 3.5. The comparison of seasonal trends with or without meteorological adjustments was removed from Figure 7. We also added a note in Section 3.3 as follows:
   *Since the sampling frequency is the only control variable in the following analysis, hereafter, unless additional demonstrations are needed to highlight the influence from other factors, we place the focus on the meteorologically adjusted mean trends over 1990-2021.*

4. The original Figure 6 was fairly complicated and difficult to understand, so we moved it to the Supplement and replaced it with a simpler plot.

These arrangements reduced some of the technical discussions in the main text, and enabled us to stay focused on the essential information.

However, we would like to keep the integrity in Figures 5 and 10, as these represent the general implications of meteorological adjustments and sampling schemes. More importantly, these figures directly address some common doubts or misperceptions about trend analysis, such as:

1. Is the trend result sensitive to different time periods (or the choice of beginning/ending years)?
2. How long does the time series need to be, in order for the trends (if any) to become detectable?
3. 30-years should be sufficiently long to wash out the sampling bias.

By having different ranges of time periods, we can better justify the importance of enhanced sampling for trend detection (i.e., the shorter the record, the larger the impact of the sampling bias; and the bias can persistently affect the trend estimate for time series as long as 40-years).

To better connect our sampling discussion to the issue of time periods, we added a discussion in the Conclusions section:

*Since the late 1980s, the challenges of quantifying global or regional-scale ozone climatologies and trends from sparsely sampled ozone profiles have been regularly revisited (Prinn, 1988; Logan 1999; Cooper et al., 2010; Saunois et al., 2012; Chang et al., 2020). While the great majority of attention has been paid to maintaining long-term operations or expanding observational networks, scant effort has been devoted to increasing the regular sampling rate. The under-appreciation of high frequency sampling might be due to a common assumption that the impact of sampling bias (along with meteorological influence) can be neutralized once the time series is sufficiently long (typically 20 or 30 years). It should be noted that the larger the data variability, the longer the data length required to detect a given trend (Weatherhead et al., 1998; Fischer et al., 2011). This paper shows that a low sampling frequency generally results in an unexpectedly larger uncertainty, which leads to suppressed statistical power and requires a much longer time period (e.g. persistence for 40 years in some cases) for free tropospheric ozone trend detection. In conclusion, we found that a regular sampling frequency of at least 3 samples per week is required to avoid most of the impact from low sampling rates.*

**The next main debatable point is the choice of ozone climatology for their so-called "adaptive sampling" strategy (D). Why choose 1980-1989 and not the entire period, or the period over which trends are computed? By choosing 1980-1989, it leaves room for a possible offset from the mean values that would be computed over the later periods, which potentially can skew the distribution of monthly mean bias on which strategy D is based upon.**

The choice of 1980-1989 is based on a practical consideration, because if we aim to mimic the improvement since 1990, we only have the 1980-1989 data available for building the

climatology (it is not realistic to have the 1980-2021 climatology available for the year 1990, because the years 1991-2021 have not yet happened).

However, it is also true that we can adjust the climatology on a regular basis to avoid potential offsets. We added a sentence in Section 3.4 as follows:

*We could also constantly update the climatology by incorporating new information from recently available samples, to better represent long-term baseline variability.*

**Other comments are minor:**
**Figure 2 and related text (page 4):**
**There is no information on the measurement uncertainty for the MLO temperature and ozone surface instruments. Instruments yielding large measurement uncertainty are likely to produce similar inconsistency between calculated trends. The authors probably assume that the total measurement uncertainty of these instruments is small enough to be taken out of the equation. If so, please state it.**

Thanks for pointing out the issue of measurement uncertainty, we added a discussion as follows (Section 2.1):

*The measurement uncertainty for the MLO records (typically ~2-4%) is assumed to be random and not explicitly taken into account in our analysis (in addition, the daily nighttime averages are expected to smooth out some measurement uncertainty). Nevertheless, if the measurement uncertainty is not random, its effect is likely to be similar to sampling bias, and their total uncertainty is expected to be propagated (and not neutralized).*

**Figure 4, lines 17-29 (note on the sampling deviation vs. sampling bias):**
**This sentence is not clear. What does the expression "insufficient number of samples to infer a monthly mean value" mean exactly? Please re-phrase/clarify.**

We revised the text as follows:

*Note that the sampling deviation associated with each daily value represents the true ozone inter-daily variability, and should not be considered to be a sampling bias; the sampling bias occurs only if we use limited samples to estimate the monthly mean value or trend.*

**Figure 7:**
**Too much information between top and bottom row. I suggest top row shows only "with meteorological adjustment" trends, and only for period 1990-2021. This way reader has a much faster access to the actual information to be used from this figure (seasonality).**

Thanks for the suggestion. The top panels were replaced with meteorologically adjusted seasonal trends over 1990-2021.

**Figure 9:**
**Because panel (b) shows a coverage rate of 2.2 samples/week, I would suggest to show a 2-samples/week example in panel (a) instead of Sunday (1 day/week).**
Thanks for the suggestion. Our intention is to show how the sampling biases (as indicated by vertical lines) are reduced by including additional samples, so instead of replacing the top panel (1-sample/week) with 2-samples/week, we replace the bottom panel (2.5 samples/week) with 1.5 samples/week. Therefore the coverage rates are more similar.

**Figure S15:**
**It would be nice here to plot the trends from ozonesonde data itself, together with the MLO subsampled data. The consistency of these two independent datasets could be demonstrated, and would provide a direct justification to apply the methods discussed in this appear to the ozonesonde launch programs, among others. On the other end, if the two datasets do not show consistent results, this would trigger (a needed) discussion on the applicability of the method discussed in this paper, for example highlighting the possible impact of measurement uncertainty and long-term stability.**

We added a new analysis in the supplementary material as follows:

*Previous trend studies of free tropospheric ozone profiles and/or columns were typically conducted without considering other covariates (apart from the basic trend model (Tiao et al., 1986; Oltmans et al., 2006)) or by only incorporating large-scale circulations, such as ENSO and QBO (Logan, 1994; Oltmans et al., 2013; Chang et al., 2022). No previous trend studies (to the best of our knowledge) have thoroughly investigated the attributions of free tropospheric ozone profile data variability to meteorological variates. Therefore, while we aim to investigate the consistency between the Hilo ozonesondes and the MLO nighttime averages (subsampled to the colocated dates), it is also desirable to consider meteorological influences on ozonesonde trends. Note that relative humidity sensors on the older sondes were not as reliable as modern sensors (Fujiwara et al., 2003), so the records before July 1991 were excluded from this analysis.*

*Since the once-per-week sampling scheme at Hilo has too few profiles to perform the resampling analysis (as we did for the MLO record in Figures 4-6), we are not able to properly quantify the improvement of trend accuracy due to covariate adjustments, so we focus on the reductions of fitted residuals (an indication of the overall fitted quality) before and after incorporating covariates. The results are shown in Figure S18:*
  - *An overall strong correlation can be found between individual Hilo ozonesondes (680 hPa) and the colocated MLO nighttime averages.*
  - *Our previous findings show that meteorological adjustments on average reduce the fitted residuals by 27% and trend uncertainty by 35% at MLO. Consistent improvements can be found at the corresponding level (680 hPa) above Hilo, by 24% and 34%, respectively, further demonstrating that the free tropospheric ozone variability can be attributed to colocated meteorological influence (i.e., dewpoint variability in this analysis).*

- *Nevertheless, highly consistent trends are still not observed between the Hilo ozonesonde (680 hPa) and the colocated MLO record. The reason behind this warrants further detailed investigation, but the combined effect of measurement uncertainty and intra-daily variability is expected to play a major role.*

**(a) correlation: Hilo v MLO**

[Figure]

**(b) Hilo trends (1991-2021)**

[Figure]

Figure S18. (a) Measurement correlation between individual Hilo ozonesondes (680 hPa) and their colocated MLO nighttime average over 1991-2021; and (b) the Hilo trend profiles, along with the MLO trends (full or colocated record), with or without the meteorological adjustments.

**Actually, stratospheric intrusions could (should?) be included among the "meteorological adjustments". There is no scientific reasons to exclude samples underlying stratospheric intrusions and yet include samples underlying other dynamical/natural variability.**

We revised the text as follows:

*Data measured during those events are highly leveraged and can be either filtered out or taken into account in meteorological adjustments (if a proper covariant can be identified) for tropospheric ozone trend analysis.*

**Raeesa Moolla**
**Overall, well written, in depth analysis, sound statistical methodological approach used.**

Thank you for the positive feedback.

**Minor comments:**
**Page 3 - Line 19 - the word 'are' is missing (averages 'are' generally much smoother)**

Fixed.

**Page 11 - section 3.4 - Is it an option to increase frequency of reading and reduce spatial variability (i.e. less sondes, but more sampling days?**

Thanks for pointing this out. Spatial variability is indeed a critical factor when studying regional trends. While spatial representativeness is also a very important topic, this study aims to present a clean analysis on how the temporal sampling can affect long-term trends. Therefore we can acknowledge how much bias can be attributed to temporal sampling schemes alone (while keeping other factors invariant).

Since the current observational network is inhomogeneously distributed and sparsely located over the globe, it is not possible to distinguish between temporal and spatial variability merely based on profile data. Fortunately, the issue of spatial representativeness will be addressed by the Chemical Reanalysis working group within the TOAR-II activity by using model evaluations (based on the assessment framework described by Miyazaki & Bowman (2017)), so the spatial variability can be better characterized. We added the relevant reference at the end of the paper when discussing spatial variability.

Miyazaki, K., & Bowman, K. (2017). Evaluation of ACCMIP ozone simulations and ozonesonde sampling biases using a satellite-based multi-constituent chemical reanalysis. Atmospheric Chemistry and Physics, 17(13), 8285-8312.

**Page 12 - line 4 - Why 2-7 days and not 2-5 days? focus has been on 2-5 days mostly, so why analyse the cost-benefit of an additional 2 days ?**

It is because we like to mimic a scenario of what happens if we have full sampling at a particular season (all 7 days per week) and once-per-week sampling at other seasons, which is summarized in Figure 7. Since we do not see consistent improvements, we then discuss the reason behind this phenomena (i.e. select bias or preferential sampling) and do not explore this sampling strategy further.

In Table 3 we choose 5 samples/week at one season and 1 sample/week at other seasons for Strategy C, because we aim to compare different sampling strategies based on similar total sample sizes (or temporal coverage). Therefore we can quantify how much improvements from

our adaptive sampling strategy, in terms of better trend accuracy and precision (greater benefit), but with similar or lower total sample sizes (lower cost).

We revised the text as follows:

*Strategy C is a mixed sampling approach in which we use once-per-week sampling for all months as the baseline, then during a particular season the frequency is increased to 2-7 samples per week (while the other seasons maintain once-per-week sampling), so we can investigate if the overall trend estimate can be improved by (partially or completely) removing specific seasonal sampling biases.*